# Comparison of Trabecular Bone Score–Adjusted Fracture Risk Assessment (TBS-FRAX) and FRAX Tools for Identification of High Fracture Risk among Taiwanese Adults Aged 50 to 90 Years with or without Prediabetes and Diabetes

**DOI:** 10.3390/medicina58121766

**Published:** 2022-11-30

**Authors:** Tzyy-Ling Chuang, Mei-Hua Chuang, Yuh-Feng Wang, Malcolm Koo

**Affiliations:** 1Department of Nuclear Medicine, Dalin Tzu Chi Hospital, Buddhist Tzu Chi Medical Foundation, Chiayi 622401, Taiwan; 2Graduate Institute of Clinical Pharmacy, Tzu Chi University, Hualien 970374, Taiwan; 3Faculty of Pharmacy, National Yang-Ming Chiao Tung University, Taipei 112304, Taiwan; 4MacKay Junior College of Medicine, Nursing, and Management, New Taipei City 11260, Taiwan; 5Department of Nuclear Medicine, Taipei Veterans General Hospital, Taipei 112021, Taiwan; 6Department of Medical Imaging and Radiological Technology, Yuanpei University of Medical Technology, Hsinchu 300102, Taiwan; 7Graduate Institute of Long-Term Care, Tzu Chi University of Science and Technology, Hualien 970302, Taiwan

**Keywords:** trabecular bone score, FRAX, fracture prediction tools, prediabetes, diabetes

## Abstract

*Background and Objectives:* The burden of osteoporosis is projected to increase. Identification and prompt intervention for osteoporotic fractures are important. Adjusting the Fracture Risk Assessment (FRAX^®^) tool with trabecular bone score (TBS) could improve risk prediction. However, little is known about whether TBS-adjusted FRAX^®^ would change the proportion of individuals qualified for osteoporosis intervention. Therefore, the aim of the present study was to compare the proportions of Taiwanese adults who qualified for intervention, according to the FRAX^®^ and TBS-adjusted FRAX^®^, with stratification by sex, age group, and glucose regulation status. *Materials and Methods:* A medical record review on adults 50–90 years who had undergone a general health examination in a regional hospital in Taiwan was conducted. FRAX^®^ and TBS-adjusted FRAX^®^ were calculated. FRAX^®^ cut-points of ≥ 20% for major osteoporotic fracture and ≥3% for hip fracture were adopted to identify individuals qualified for osteoporosis intervention. Individuals were classified as prediabetes and diabetes if their HbA1c was 5.7–6.4% and >6.4%, respectively. *Results:* A total of 8098 individuals with a mean age of 61.0 years were included. The proportion of men qualified for intervention for hip fracture was significantly lower according to TBS-adjusted FRAX^®^ (17.2%) compared with FRAX^®^ (20.7%) (*p* < 0.001), with a similar pattern across all three age groups and in those with prediabetes. In contrast, the proportion of women qualified for intervention for major osteoporotic fracture was significantly higher according to TBS-adjusted FRAX^®^ (4.6%) compared with FRAX^®^ (3.7%) (*p* < 0.001), particularly among those with prediabetes 60–69 years. *Conclusions:* TBS-adjusted FRAX^®^ led to small but significant changes in the proportions of individuals qualified for intervention in specific age groups and glucose regulation status.

## 1. Introduction

Osteoporosis is a common condition characterized by low bone mass and deterioration of bone tissue, leading to bone fragility and consequent increased risk of fracture. It remains a silent disease that is underdiagnosed and undertreated until fractures occur, which are associated with disability, dependence, and premature death [1]. Conventionally, osteoporosis is diagnosed by measuring the areal bone mineral density (BMD) using dual-energy X-ray absorptiometry (DXA). As defined by the World Health Organization (WHO), osteoporosis is present when the BMD is 2.5 standard deviations or more below the average value for young healthy women [2]. However, BMD does not completely reflect fracture risk because bone tissue material properties also play a role in the ability of a whole bone to resist stress fracture [3,4].

A novel development in the non-invasive measurement of trabecular microarchitecture is the trabecular bone score (TBS) introduced in 2008 [5]. TBS is obtained indirectly based on the use of experimental variograms of two-dimensional projection images on gray-level texture derived from lumbar spine DXA scanning [6]. TBS has been shown to predict incident hip and major osteoporotic fractures in many studies, independent of areal BMD and other clinical risk factors [7,8,9,10]. Clinically, TBS can be used to assess fracture risk and determine treatment initiation in conjunction with the Fracture Risk Assessment (FRAX^®^) tool to calculate the probability of fracture [11].

The FRAX^®^ tool was developed and released by the World Health Organization Collaborating Centre for Metabolic Bone Diseases at Sheffield, United Kingdom, in 2008 to identify individuals at high risk of a fracture using country-specific models. A worldwide usage report of FRAX^®^ showed that more than 2.3 million calculations were sourced from 173 countries over a one-year period from 1 May 2012 [12]. The tool can be used to estimate the 10-year major osteoporotic fracture probability and 10-year hip fracture probability based on eight dichotomous risk factors, namely sex, prior fragility fracture, parental history of hip fracture, current tobacco smoking, long-term use of glucocorticoids, rheumatoid arthritis, other causes of secondary osteoporosis, and excess alcohol consumption, and three continuous variables, including age, body weight, and height. In addition, BMD at the femoral neck can be entered as an optional variable [13]. A systematic review of intervention thresholds based on FRAX^®^ concluded that FRAX^®^ could identify high-risk individuals more effectively than the use of BMD, but the intervention thresholds should be country-specific [14].

With the availability of TBS measurement, TBS can be used to adjust the FRAX^®^ score, with the aim of improving fracture risk prediction. Studies have shown that the addition of TBS could improve FRAX^®^ assessments for predicting major osteoporotic fracture and hip fracture in some populations, including Japanese men [15], Chinese men and women [16], Canadians with diabetes [17], and United States men [18]. Nevertheless, other studies in Australian men [19], Sri Lankan women [20], and Korean women [21] showed no significant improvements in the predictive value for major osteoporotic fracture and hip fracture between unadjusted and TBS-FRAX^®^ scores. Overall, a meta-analysis of individual-level data from 17,809 men and women between the ages of 40 and 90 years in 14 prospective population-based cohorts concluded that TBS was a significant predictor of fracture risk independently of FRAX^®^, which supported the use of TBS as a post hoc adjunct to risk assessment with the FRAX^®^ score. A small increase in the gradient of risk (hazard ratio per 1 standard deviation (SD) change in risk variable in direction of increased risk) for major osteoporotic fracture from 1.70 (95% confidence interval (CI) 1.60–1.81) for FRAX^®^ alone to 1.76 (95% CI 1.65–1.87) for TBS-FRAX^®^ was observed [22].

In addition to the quantification of fracture risk, an objective of the FRAX^®^ tool is to help in identifying individuals who would be qualified to receive treatment for osteoporosis [23]. According to the current clinician’s guide published by the Bone Health and Osteoporosis Foundation (formerly the National Osteoporosis Foundation), for postmenopausal women and men aged ≥ 50 years, one of the indications in qualifying for pharmacologic intervention is individuals with FRAX^®^ 10-year risk scores of ≥3% for hip fracture or ≥20% for major osteoporotic fracture and T-score between −1.0 and −2.5 at the femoral neck or total hip [24]. A cross-sectional study compared the proportion of 413 older adults who should be treated for osteoporosis according to the FRAX^®^ and TBS-FRAX^®^. Overall, the proportion of individuals with a risk of major osteoporotic fracture above the threshold value of therapeutic intervention (i.e., ≥20%) was found to be similar using FRAX^®^ (24.7%) and TBS-FRAX^®^ (25.4%). However, when data were stratified into different age groups, it was found that the proportion was significantly higher according to the TBS-FRAX^®^ compared with FRAX^®^ among adults 60–70 years (38.3% vs. 30.9%) and 70–80 years (31.2% vs. 26.6%) [25].

The burden of type 2 diabetes is increasing globally. It was estimated that in 2017, 462 million individuals were affected by it, corresponding to 6.3% of the world’s population [26]. There is substantial evidence that diabetes is associated with an increased risk of osteoporotic fractures [27,28]. However, BMD and FRAX^®^ scores are known to underestimate the risk of fractures in individuals with type 2 diabetes [17,29]. A meta-analysis of 12 studies involving 40,508 adults revealed that individuals with diabetes and prediabetes had significantly lower TBS compared with normal individuals [30]. Similarly, a retrospective study of 169 postmenopausal women with type 2 diabetes showed that only TBS-FRAX^®^ score, and not BMD or FRAX^®^ score, were significantly associated with vertebral fractures [31]. Taken together, these studies suggested that TBS and TBS-FRAX^®^ could be used as supplementary tools to discriminate osteoporotic fractures in individuals with type 2 diabetes. Nevertheless, previous studies have not evaluated the consistency between the FRAX^®^ and TBS-FRAX^®^ in identifying individuals qualified for osteoporosis intervention, especially separately in men and women of different age groups, as well as in individuals with normoglycemia, prediabetes, and diabetes. Therefore, the aim of this study was to compare the proportions of Taiwanese adults at high fracture risk that qualified for intervention, according to the FRAX^®^ and TBS-FRAX^®^, with stratification by sex, age group, and glucose regulation status.

## 2. Materials and Methods

### 2.1. Study Design and Study Population

This retrospective medical record review study was conducted in a regional teaching hospital in southern Taiwan. Figure 1 shows the flowchart of the study. Individuals who had undergone a general health examination from 1 June 2014 to 30 July 2020 were reviewed for eligibility. In Taiwan, all residents are eligible to receive a free comprehensive health examination once every three years for those who are 40 to 64 years old and once a year for those aged 65 and older. The study protocol was approved by the institutional review board of the study hospital (IRB no. B11001010), and the informed consent requirements were waived due to the use of de-identified medical records.

The exclusion criteria were individuals with an age of <50 years or >90 years; lack of data on TBS, BMD, or glycated hemoglobin (HbA1c); and a history of bone fracture. In addition, individuals with a BMI < 15 or >37 kg/m^2^ were excluded according to the guidelines of the manufacturer of the TBS software.

### 2.2. Assessment of Fracture Risk by Using FRAX^®^ and TBS-FRAX^®^

Lumbar spine BMD was measured from L1 to L4 using a Discovery Wi DXA system (Hologic Inc., Marlborough, MA, USA) according to the standard operation procedures of the Department of Nuclear Medicine at the study hospital. TBS was retrospectively calculated from L1 to L4 from the same DXA scans using TBS iNsight^®^ software (version 3.0.2.0; Medimaps Group, Geneva, Switzerland).

FRAX^®^ and TBS-FRAX^®^ used to estimate the 10-year probability of major osteoporotic fracture and hip fracture were calculated on the FRAX^®^ website (https://www.sheffield.ac.uk/FRAX/tool.aspx?country=26) (accessed on 1 November 2022) based on the algorithm for Taiwanese adults. Information on left femoral neck BMD, age, sex, body weight, height, history of prior osteoporotic fracture, parental history of hip fragility fracture, current smoking, long-term use of oral glucocorticoids, rheumatoid arthritis, alcohol consumption (≥3 units/day), and other secondary causes of osteoporosis were input into the FRAX^®^ website for each individual. The use of the left side data was based on the assumption that that the left side is often the non-dominant side and that the non-dominant side is exposed to less impact, resulting in lower BMD. A study on 2372 Caucasian women concluded that the extra scan time, cost, and radiation dose associated with bilateral BMD measurements might not be justified [32]. Double data entry by two trained research assistants was used to ensure the accuracy of the data acquisition process. FRAX^®^ cut-points of ≥20% for major osteoporotic fracture and ≥3% for hip fracture were adopted to identify individuals at high risk for fracture qualified for osteoporosis intervention [33].

### 2.3. Type 2 Diabetes and Prediabetes and Other Demographic Information

Individuals were classified as type 2 prediabetes if their HbA1c was 5.7% to 6.4% and as type 2 diabetes if their HbA1c was greater than 6.4% [34]. Basic demographic and clinical information was obtained from the records of the general health examination.

### 2.4. Statistical Analysis

All statistical analyses were performed using IBM SPSS Statistics for Windows, Version 25.0.0.2 (IBM Corp., Armonk, NY, USA). Continuous variables were summarized as mean with standard deviation (SD). The differences between men and women for the basic demographic and clinical variables were compared using the independent *t*-test. One-way analysis of variance with Bonferroni adjustment was used to compare the means of lumbar spine BMD, hip neck femoral BMD, and hip total BMD between the normoglycemia, prediabetes, and diabetes group.

In addition, the non-parametric McNemar’s test for paired binomial data was used to assess the discordance in the proportions between FRAX^®^ and TBS-FRAX^®^ tools in classified individuals as high risk for major osteoporotic fracture and hip fracture. A value ≥ 20% and ≥3% was defined as high risk for major osteoporotic fractures and hip fractures, respectively [32].

## 3. Results

A total of 8098 individuals were included in the present study. The mean age was 61.0 years (SD 7.2 years), and 71.7% of them were women. The demographic and clinical characteristics of the individuals, stratified by sex and glucose regulation status, are shown in Table 1. All 18 variables were significantly different between men and women. When stratified by glucose regulation status, only albumin was not significantly different between men and women in the diabetes group; fasting blood glucose was not significantly different between men and women in the prediabetes group; and age, body mass index, systolic blood pressure, low-density lipoprotein cholesterol, fasting blood glucose, and albumin were not significantly different between men and women in the diabetes group.

Figure 2 shows the scatterplots of lumbar spine BMD, hip neck femoral BMD, and hip total BMD in the normoglycemia, prediabetes, and diabetes group. The lumbar spine BMD and hip neck femoral BMD of the diabetes group were significantly higher compared with those of the other two groups. The hip neck femoral BMD of the diabetes group was significantly higher than that of the prediabetes group, which in turn was significantly higher than that of the normoglycemia group.

The proportions of men and women qualified for intervention for major osteoporotic fracture (i.e., FRAX^®^ and TBS-FRAX^®^ ≥ 20%) and hip fracture (i.e., FRAX^®^ and TBS-FRAX^®^ ≥ 3%) are shown in Table 2. The proportion of men qualified for intervention for hip fracture was significantly lower according to TBS-FRAX^®^ (*p* < 0.001). In contrast, the proportion of women qualified for intervention for major osteoporotic fracture was significantly higher according to TBS-FRAX^®^ (*p* < 0.001).

When the analyses were stratified by age group, a similar pattern was observed in men in the age groups of 50 to 59 years (*p* < 0.001) and 60 to 69 years (*p* < 0.001). In women, a similar pattern was observed in the age groups of 60 to 69 years (*p* < 0.001) and 70 to 90 years (*p* = 0.029). Moreover, the proportion of women qualified for intervention for hip fracture was significantly lower according to the TBS-FRAX^®^ (*p* = 0.033).

Table 3 shows the proportions of men in different age groups qualified for intervention for major osteoporotic fracture (i.e., FRAX^®^ and TBS-FRAX^®^ ≥ 20%) and hip fracture (i.e., FRAX^®^ and TBS-FRAX^®^ ≥ 3%), stratified by glucose regulation status. For men in the age group of 50 to 59 years, the proportion qualified for intervention for hip fracture was significantly lower according to the TBS-FRAX^®^ in both the normoglycemia group and prediabetes group (*p* < 0.001).

For men in the age group of 60 to 69 years, the proportions qualified for intervention for hip fracture were significantly lower according to the TBS-FRAX^®^ in the normoglycemia, prediabetes, and diabetes groups (Table 3).

For men in the age group of 70 to 90 years, none of the proportions qualified for intervention for major osteoporotic fracture or hip fracture were significantly different between the FRAX^®^ and TBS-FRAX^®^ for the normoglycemia, prediabetes, or diabetes groups (Table 3).

Table 4 shows the proportions of women in different age groups qualified for intervention for major osteoporotic fracture (i.e., FRAX^®^ and TBS-FRAX^®^ ≥ 20%) and hip fracture (i.e., FRAX^®^ and TBS-FRAX^®^ ≥ 3%), stratified by glucose regulation status. For women in the age group of 50 to 59 years, only the proportion qualified for intervention for hip fracture was significantly lower according to the TBS-FRAX^®^ in the normoglycemia group (*p* = 0.020).

For women in the age group of 60 to 69 years, the proportions qualified for intervention for major osteoporotic fracture were significantly higher according to the TBS-FRAX^®^ in the normoglycemia group and the prediabetes group. In addition, the proportion of women above the treatment threshold for hip fracture was significantly higher according to TBS-FRAX^®^ in the prediabetes group (*p* = 0.044) (Table 4).

Finally, for women in the age group of 70 to 90 years, none of the proportions qualified for intervention for major osteoporotic fracture or hip fracture were significantly different between the FRAX^®^ and TBS-FRAX^®^ classification for the normoglycemia, prediabetes, or diabetes groups (Table 4).

## 4. Discussion

In this study of 8098 middle-aged to older adults identified from the medical records of general health examinations conducted in a regional teaching hospital in southern Taiwan, we compared the consistency of the FRAX^®^ and TBS-FRAX^®^ in determining the proportions of men and women at high risk of fracture that qualified for osteoporosis intervention. Comparisons were made between the two tools for major osteoporotic fracture and hip fracture, separately for men and women, and for three different age groups. In addition, comparisons between two tools were made in individuals with normal glucose regulation, prediabetes, and diabetes. Individuals that qualified for intervention when their results of FRAX^®^ and TBS-FRAX^®^ were ≥3% for hip fracture or ≥20% for major osteoporotic fracture were considered as high risk of fracture. This is also one of the criteria for initiating pharmacological interventions for osteoporosis in postmenopausal women and men 50 years of age according to the National Osteoporosis Foundation guidelines [35].

Overall, our results showed that the use of TBS to adjust for FRAX^®^ led to small but significant changes in the proportions of individuals with high fracture risk qualified for intervention in certain age groups and also differently between men and women. Compared with the FRAX^®^, the TBS-FRAX^®^ showed a significantly lower proportion of men at high risk for hip fracture but a significantly higher proportion of women at high risk for major osteoporotic fracture. In other words, using the TBS-FRAX^®^ compared with FRAX^®^, fewer men and more women would be qualified for pharmacological intervention for osteoporosis. In addition, when stratified by age group, the changes appeared to be consistent among men aged 50 to 69 years and among women aged 60 to 90 years.

In a study of 413 French adults (84.5% women) hospitalized for nonvertebral fractures, the FRAX^®^ and TBS-FRAX^®^ scores were calculated without considering the current fracture. It was found that the proportions of individuals with a risk of major osteoporotic fracture ≥ 20% and a risk of hip fracture ≥ 3%, before the current fracture, were similar between FRAX^®^ and TBS-FRAX^®^ (24.7% vs. 25.4% and 60.0% vs. 59.6%, respectively). However, the proportion of individuals identified with a risk of major osteoporotic fracture above the intervention threshold was significantly higher according to the TBS-FRAX^®^ scores for the age group 60 to 69 years (38.3% vs. 30.9%) and 70 to 79 years (31.2% vs. 26.6%) [25]. It should be noted that the participants in the French study consisted of those hospitalized for nonvertebral fractures, whereas those in the present study were made up of relatively healthy individuals. Therefore, it is not surprising to find that the magnitude of the proportion of individuals qualified for osteoporosis intervention was higher than that observed in the present study. While the French study reported that TBS-FRAX^®^ could identify a higher number of older adults qualified for intervention, our study further showed that this difference existed only in women 60 years and older.

Previous research has shown that type 2 diabetes was associated with an increase in bone fragility but also paradoxically with a higher BMD [36,37]. Our results also showed that BMD was significantly higher in individuals with diabetes compared to those with normoglycemia. TBS and TBS-FRAX^®^ have been considered as complementary tools to discriminate osteoporotic fractures in individuals with type 2 diabetes [31]. A study on 4100 Vietnamese adults reported that women, but not men, with type 2 diabetes and prediabetes had lower TBS than individuals without diabetes [38], suggesting that degradation of bone microarchitecture could occur in early stages of impaired glucose regulation. However, to our knowledge, no studies have assessed whether the FRAX^®^ and TBS-FRAX^®^ would identify a comparable proportion of individuals who would be qualified for receiving osteoporosis intervention in those with prediabetes and diabetes. Therefore, the present study evaluated if individuals with impaired glucose regulation would exhibit a different pattern according to the FRAX^®^ and TBS-FRAX^®^ tools. First, no significant differences were observed between the two tools when applied to the oldest age group (70 to 90 years) among individuals with prediabetes and diabetes.

Second, there were some significant differences in the youngest age group (50 to 59 years). In men, the proportion at high risk for hip fracture qualified for intervention was significantly lower according to the TBS-FRAX^®^ in those with normoglycemia and prediabetes. However, no significant differences between the two indexes were observed in the diabetes groups.

Third, the 60 to 69 years age group showed the largest number of significant differences between the two tools. In men, the proportion at high risk of hip fracture qualified for intervention was significantly lower according to the TBS-FRAX^®^ in those with normoglycemia, prediabetes, and diabetes. In women, the proportions at high risk of major osteoporotic fracture and hip fracture qualified for intervention were significantly higher according to the TBS-FRAX^®^ in those with normoglycemia and prediabetes. However, no significant differences between the two tools were observed in those with diabetes. In summary, in individuals with diabetes, the use of TBS-FRAX^®^ did not significantly change the proportion qualified for osteoporosis intervention except in men aged 60 to 69 years.

Our findings in individuals with diabetes are consistent with those obtained in studies that evaluated the prediction ability of the TBS-FRAX^®^. In a case-control study of 80 postmenopausal women with type 2 diabetes, no significant differences were observed between women with or without diabetes in the risk of major osteoporotic fracture and hip fracture assessed by FRAX^®^ or TBS-FRAX^®^ [39]. In contrast, the results from a cohort study based on the Manitoba BMD Registry showed that adjusting FRAX^®^ with TBS was associated with a moderate and predominantly upward reclassification of 4.1% and 5.7% for major osteoporotic fracture (cut-off of 20%) and hip fracture (cut-off of 3%), respectively, in individuals with diabetes [17]. As the results of the Manitoba study did not present separately for men and women and in different age groups, they could not be compared directly with the present study. Furthermore, in the observational Geelong Osteoporosis Study of 1069 Australian adults, adjusting FRAX^®^ with TBS resulted in higher scores in individuals with diabetes to a greater extent than in those with normoglycemia or impaired fasting glucose. The differences were particularly prominent in women under the age of 65 years [40]. In short, the clinical utility of adjusting FRAX^®^ with TBS in predicting fracture risk in individuals with impaired glucose regulations are still equivocal. Further studies are warranted to clarify the predictive significance of the TBS-FRAX^®^, particularly in the early stage of glucose impairment.

This study had several limitations. First, the study used a cross-sectional study design based on review of medical records from a general health examination conducted in a regional teaching hospital. The study population was not a nationally representative sample. Second, we compared the differences in proportions of individuals at high risk of major osteoporotic fracture and hip fracture qualified for intervention according to the FRAX^®^ and TBS-FRAX^®^ scores. Longitudinal studies are needed to further examine whether adjusting FRAX^®^ with TBS can significantly improve the accuracy of predicting future fracture risk, particularly in individuals with prediabetes. Third, information on the type of diabetes and whether anti-fracture medication was used is unavailable from the medical records of the general health examination. Fourth, the relatively large number of comparisons conducted in this study might increase the risk of type I error. Nevertheless, for the purpose of preventive identification, it may be more prudent to err on the side of false positives.

Despite the aforementioned limitations, this study has some strengths. First, our large study sample included both men and women over a broad range of adult ages. This is important as men are relatively understudied in osteoporosis research. Second, all our TBS measurements were conducted with the same densitometer and TBS algorithm, which eliminated potential differences in the results introduced by densitometers [41]. Third, the present study provided comprehensive comparisons of the risk of major osteoporotic fracture and hip fracture assessed by FRAX^®^ or TBS-FRAX^®^ separately for men and women of different age groups, as well as in individuals with normoglycemia, prediabetes, and diabetes.

## 5. Conclusions

Due to increased life expectancy and an aging population, the burden of osteoporosis and fragility fractures is projected to continue to increase in the next decade [42]. Therefore, preventive identification and prompt intervention for the risk of osteoporotic fractures are important and timely issues. Findings from this study showed that the use of TBS-FRAX^®^, instead of FRAX^®^, could change the proportion of middle-aged and older adults qualified for osteoporosis treatment. Specifically, a lower number in men aged 50 to 69 years and a higher number in women aged 60 years and over were found to reach the threshold for intervention with the use of the TBS-FRAX^®^. Moreover, a similar but slightly different pattern was observed in individuals with prediabetes but not diabetes.

## Figures and Tables

**Figure 1 medicina-58-01766-f001:**
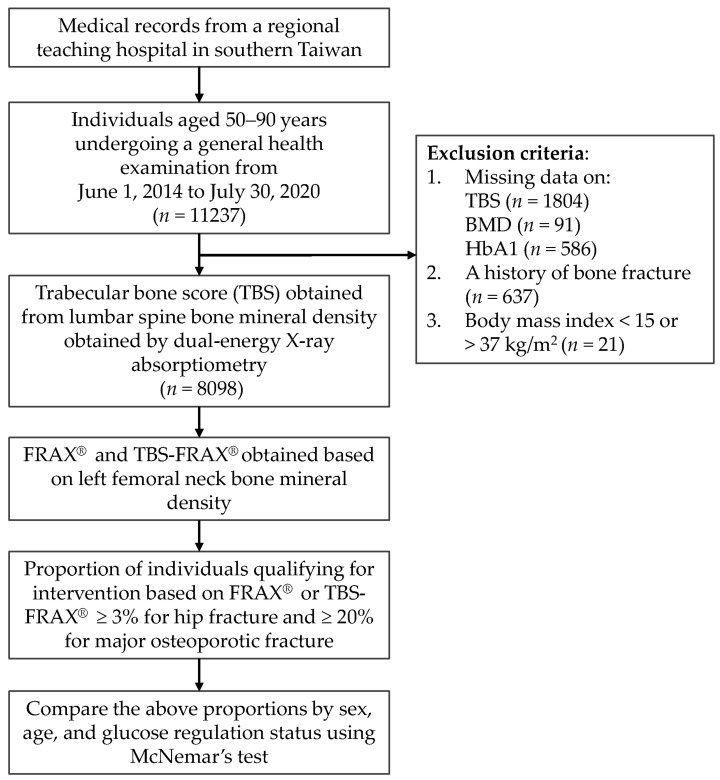
Study flowchart.

**Figure 2 medicina-58-01766-f002:**
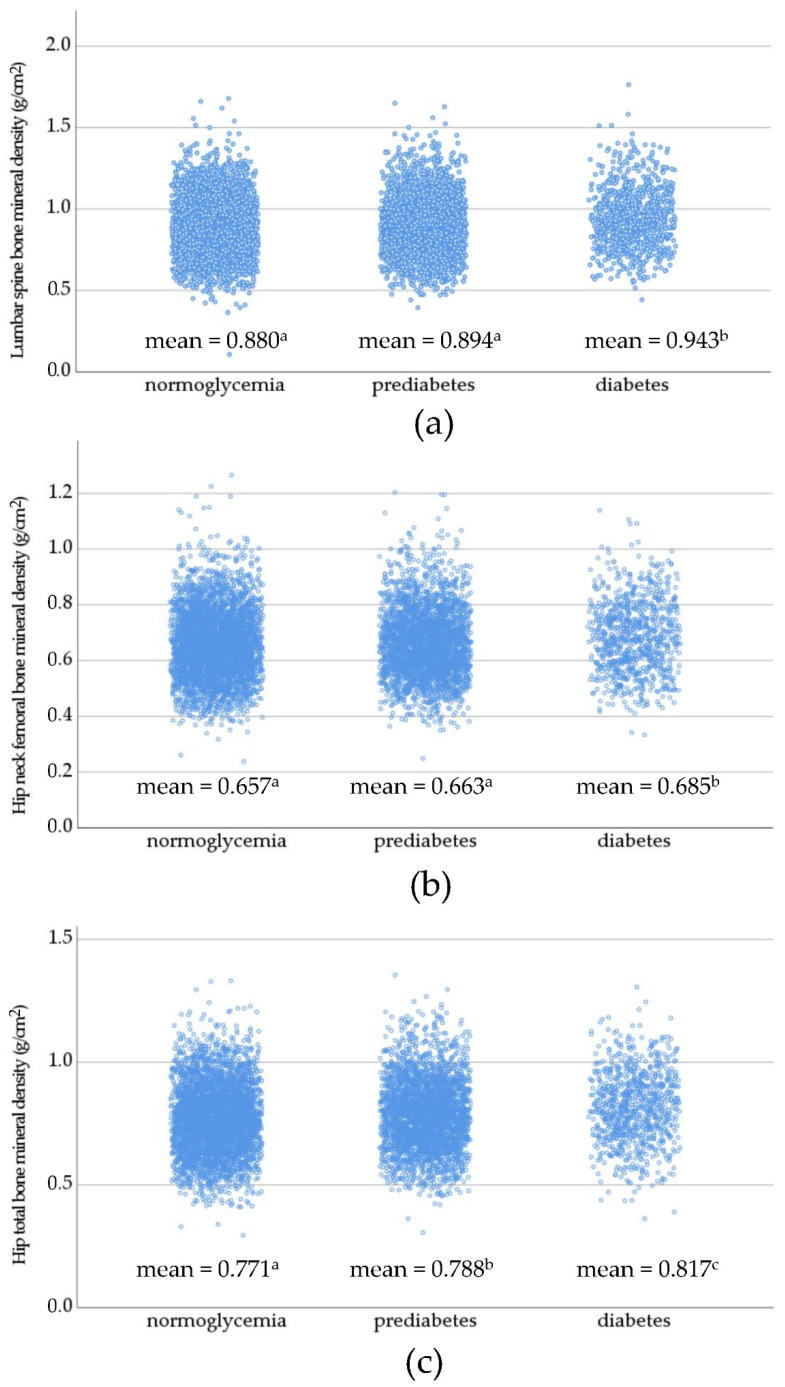
Scatterplot of lumbar spine bone mineral density (**a**); hip neck femoral bone mineral density (**b**); and hip total bone mineral density (**c**) in the normoglycemia, prediabetes, and diabetes group. The means were compared using analysis of variance with Bonferroni adjustment for pairwise comparisons. Different superscript denotes statistically significant differences between group means (*p* < 0.05).

**Table 1 medicina-58-01766-t001:** The demographic and clinical characteristics of the individuals (*n* = 8098).

Variable	All (*n* = 8098)			Glucose Regulation Status								
				Normoglycemia *n* = 4126 (51%)			Prediabetes *n* = 3078 (38%)			Diabetes *n* = 894 (11%)		
	Men (*n* = 2959)	Women (*n* = 5139)	*p*	Men (*n* = 1514)	Women (*n* = 2612)	*p*	Men (*n* = 1045)	Women (*n* = 2033)	*p*	Men (*n* = 400)	Women (*n* = 494)	*p*
Age (years)	61.72 (7.52)	60.69 (7.05)	<0.001	60.62 (7.23)	59.46 (6.91)	<0.001	62.60 (7.58)	61.55 (6.91)	<0.001	63.56 (7.83)	63.63 (7.05)	0.884
BMI (kg/m^2^)	24.69 (3.04)	23.62 (3.26)	<0.001	24.21 (2.86)	22.92 (2.96)	<0.001	24.96 (3.12)	24.11 (3.28)	<0.001	25.77 (3.10)	25.34 (3.68)	0.058
SBP (mmHg)	132.43 (19.79)	128.19 (21.06)	<0.001	131.53 (19.81)	125.86 (21.11)	<0.001	132.57 (19.57)	129.87 (20.80)	<0.001	135.5 (20.06)	133.56 (20.27)	0.155
DBP (mmHg)	79.04 (10.89)	71.94 (11.08)	<0.001	79.01 (11.25)	71.26 (11.20)	<0.001	78.90 (10.35)	72.61 (10.94)	<0.001	79.55 (10.89)	72.76 (10.81)	<0.001
Cholesterol (mg/dL)	177.29 (36.89)	192.68 (36.78)	<0.001	176.10 (35.35)	191.38 (35.70)	<0.001	180.97 (36.71)	197.00 (36.99)	<0.001	172.20 (41.91)	181.81 (38.82)	<0.001
Triglycerides (mg/dL)	124.96 (83.13)	112.3 (60.72)	<0.001	115.4 (85.72)	101.54 (54.33)	<0.001	129.63 (75.8)	120.29 (61.57)	<0.001	148.96 (85.82)	136.30 (75.59)	0.021
HDLC (mg/dL)	43.07 (11.85)	52.52 (14.38)	<0.001	44.33 (12.27)	54.71 (14.79)	<0.001	42.66 (11.39)	50.81 (13.84)	<0.001	39.40 (10.52)	47.95 (12.23)	<0.001
LDLC (mg/dL)	114.74 (31.40)	120.89 (31.88)	<0.001	113.95 (29.94)	119.24 (30.83)	<0.001	117.74 (31.32)	125.37 (32.17)	<0.001	109.89 (35.93)	111.14 (33.23)	0.593
Fasting blood glucose (mg/dL)	109.28 (24.85)	106.58 (23.19)	<0.001	99.17 (8.45)	98.03 (8.73)	<0.001	108.15 (14.21)	106.22 (13.11)	0.222	150.48 (41.06)	153.20 (43.82)	0.345
Albumin (g/dL)	4.36 (0.31)	4.34 (0.31)	0.001	4.35 (0.30)	4.35 (0.32)	0.437	4.37 (0.32)	4.33 (0.30)	0.001	4.38 (0.32)	4.34 (0.33)	0.075
ALP (IU/L)	75.46 (22.86)	81.91 (23.68)	<0.001	75.84 (22.70)	81.51 (23.61)	<0.001	75.39 (23.29)	82.38 (23.78)	<0.001	74.17 (22.35)	82.11 (23.63)	<0.001
eGFR (mL/min/1.73 m^2^)	80.75 (16.52)	109.47 (23.72)	<0.001	83.25 (16.02)	112.17 (23.78)	<0.001	79.03 (16.22)	107.03 (22.76)	<0.001	75.82 (17.47)	105.16 (25.52)	<0.001
Lumbar spine BMD (g/cm^2^)	0.98 (0.16)	0.84 (0.14)	<0.001	0.96 (0.15)	0.84 (0.14)	<0.001	0.99 (0.16)	0.84 (0.14)	<0.001	1.04 (0.17)	0.87 (0.14)	<0.001
Hip neck femoral BMD (g/cm^2^)	0.72 (0.11)	0.63 (0.10)	<0.001	0.71 (0.11)	0.62 (0.10)	<0.001	0.72 (0.11)	0.63 (0.10)	<0.001	0.74 (0.11)	0.64 (0.11)	<0.001
Hip total BMD (g/cm^2^)	0.86 (0.13)	0.74 (0.12)	<0.001	0.84 (0.12)	0.73 (0.12)	<0.001	0.86 (0.13)	0.75 (0.12)	<0.001	0.88 (0.12)	0.76 (0.13)	<0.001
TBS	1.38 (0.09)	1.31 (0.09)	<0.001	1.38 (0.08)	1.31 (0.09)	<0.001	1.37 (0.09)	1.30 (0.09)	<0.001	1.38 (0.09)	1.29 (0.09)	<0.001

ALP: alkaline phosphatase; BMD: bone mineral density; BMI: body mass index; DBP: diastolic blood pressure; eGFR: estimated glomerular filtration rate; HDLC: high-density lipoprotein cholesterol; LDLC: low-density lipoprotein cholesterol; SBP: systolic blood pressure; TBS: trabecular bone score. All values are mean and standard deviation unless stated otherwise.

**Table 2 medicina-58-01766-t002:** The proportions of men and women at high risk of major osteoporotic fracture (≥20%) and hip fracture (≥3%) according to the FRAX^®^ and TBS-FRAX^®^, with stratification by age group.

Variable	All		Age Group (Years)					
			50–59		60–69		70–90	
	% qualified for intervention	*p*	% qualified for intervention	*p*	% qualified for intervention	*p*	% qualified for intervention	*p*
**Men**	(*n* = 2959)		(*n* = 1336)		(*n* = 1165)		(*n* = 458)	
**Major osteoporotic fracture risk**		>0.999		NC		NC		>0.999
FRAX^®^	0.1		0		0		0.7	
TBS-FRAX^®^	0.1		0		0		0.9	
**Hip fracture risk**		<0.001		<0.001		<0.001		0.664
FRAX^®^	20.7		8.2		22.7		51.7	
TBS-FRAX^®^	17.2		5.4		17.4		51.1	
**Women**	(*n* = 5139)		(*n* = 2571)		(*n* = 1983)		(*n* = 585)	
**Major osteoporotic fracture risk**		<0.001		0.250		<0.001		0.029
FRAX^®^	3.7		0.1		2.7		22.7	
TBS-FRAX^®^	4.6		0.2		4.3		25.0	
**Hip fracture risk**		0.800		0.033		0.371		0.327
FRAX^®^	31.5		10.5		43.3		83.4	
TBS-FRAX^®^	31.4		9.6		43.9		84.4	

TBS: trabecular bone score; FRAX^®^: Fracture Risk Assessment; NC: not calculable. *p* values were obtained using McNemar’s test.

**Table 3 medicina-58-01766-t003:** The proportions of men in different age groups at high risk of major osteoporotic fracture (≥20%) and hip fracture (≥3%) according to the FRAX^®^ and TBS-FRAX^®^, with stratification by glucose regulation status.

Variable	Glucose Regulation Status					
	Normoglycemia		Prediabetes		Diabetes	
	% qualified for intervention	*p*	% qualified for intervention	*p*	% qualified for intervention	*p*
**50 to 59** **years of age**	(*n* = 769)		(*n* = 428)		(*n* = 139)	
**Major osteoporotic fracture risk**		NC		NC		NC
FRAX^®^	0		0		0	
TBS-FRAX^®^	0		0		0	
**Hip fracture risk**		<0.001		<0.001		0.250
FRAX^®^	9.1		7.7		5.0	
TBS-FRAX^®^	6.0		5.1		2.9	
**60 to 6** **9 years of age**	(*n* = 563)		(*n* = 426)		(*n* = 176)	
**Major osteoporotic fracture risk**		NC		NC		NC
FRAX^®^	0		0		0	
TBS-FRAX^®^	0		0		0	
**Hip fracture risk**		<0.001		0.011		0.006
FRAX^®^	26.3		18.8		21.0	
TBS-FRAX^®^	19.4		15.7		15.3	
**70 to** **90 years of age**	(*n* = 182)		(*n* = 191)		(*n* = 85)	
**Major osteoporotic fracture risk**		>0.999		>0.999		NC
FRAX^®^	0.5		1.0		0	
TBS-FRAX^®^	1.1		1.0		0	
**Hip fracture risk**		>0.999		>0.999		0.219
FRAX^®^	48.9		56.0		48.2	
TBS-FRAX^®^	49.5		56.0		43.5	

TBS: trabecular bone score; FRAX^®^: Fracture Risk Assessment; NC: not calculable. *p* values were obtained using McNemar’s test.

**Table 4 medicina-58-01766-t004:** The proportions of women in different age groups at high risk of major osteoporotic fracture (≥20%) and hip fracture (≥3%) according to the FRAX^®^ and TBS-FRAX^®^, with stratification by glucose regulation status.

Variable	Glucose Regulation Status					
	normoglycemia		prediabetes		diabetes	
	% qualified for intervention	*p*	% qualified for intervention	*p*	% qualified for intervention	*p*
**50 to 59** **years of age**	(*n* = 1512)		(*n* = 905)		(*n* = 154)	
**Major osteoporotic fracture risk**		0.500		NC		NC
FRAX^®^	0.1		0		0	
TBS-FRAX^®^	0.3		0.1		0	
**Hip fracture risk**		0.020		0.500		>0.999
FRAX^®^	11.0		10.2		7.8	
TBS-FRAX^®^	9.8		9.6		8.4	
**60 to 6** **9 years of age**	(*n* = 854)		(*n* = 880)		(*n* = 249)	
**Major osteoporotic fracture risk**		<0.001		0.002		>0.999
FRAX^®^	2.7		2.4		3.6	
TBS-FRAX^®^	4.9		3.9		4.0	
**Hip fracture risk**		0.704		0.044		>0.999
FRAX^®^	48.1		40.1		38.2	
TBS-FRAX^®^	47.7		41.8		38.2	
**70 to** **90 years of age**	(*n* = 246)		(*n* = 248)		(*n* = 91)	
**Major osteoporotic fracture risk**		>0.999		0.092		0.125
FRAX^®^	29.7		17.7		17.6	
TBS-FRAX^®^	30.1		20.6		23.1	
**Hip fracture risk**		0.774		0.754		0.625
FRAX^®^	85.8		83.5		76.9	
TBS-FRAX^®^	86.6		84.3		79.1	

TBS: trabecular bone score; FRAX^®^: Fracture Risk Assessment; NC: not calculable. *p* values were obtained using McNemar’s test.

## Data Availability

The data used to support the findings of this study are available from the corresponding author upon request.

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
