# Peer review of "Comparison of Trabecular Bone Score–Adjusted Fracture Risk Assessment (TBS-FRAX) and FRAX Tools for Identification of High Fracture Risk among Taiwanese Adults Aged 50 to 90 Years with or without Prediabetes and Diabetes"

_medicina, 2022, doi:10.3390/medicina58121766_

Round 1
Reviewer 1 Report
This manuscript is considered additional fracture risk considering trabecular bone score (TBS), which is the evaluation of trabecular microarchitecture. After a review of your manuscript, I am pleased to inform you that I think it is worth publishing in this journal.
Author Response
Reviewer 1, comment 1:
This manuscript is considered additional fracture risk considering trabecular bone score (TBS), which is the evaluation of trabecular microarchitecture. After a review of your manuscript, I am pleased to inform you that I think it is worth publishing in this journal.
Response to Reviewer 1, comment 1:
We greatly appreciate the positive comment and encouragement from the reviewer.
Reviewer 2 Report
Thank you for the opportunity to review this manuscript. I have previously reviewed this manuscript for another journal. The authors have responded to some of my previous comments, but not all of them. Please see below for the remaining comments from my previous review:
This is a well written manuscript that presents some interesting results. However, it requires some editing prior to acceptance for publication. In particular, the most interesting message of this manuscript doesn’t stand out, which is the effect of adjusting FRAX using TBS for participants with prediabetes or diabetes. The manuscript can be edited to highlight these observations more clearly.
Abstract:
1. The results presented in the abstract are unclear. Could the authors please provide some values in the abstract to show the results more clearly?
Methods:
2. A participant flow chart would be useful for this study, to show the exclusions and how many participants were included in the final study.
3. Were there any participants taking anti-fracture medication? Were these included or excluded from the study?
4. Additional information is needed about how the “other” information included in the FRAX algorithm was obtained. Would it be that medical records were examined and the appropriate data extracted? Was the data largely complete? Or did some participants have missing data?
5. Were there any participants with type 1 diabetes? Were the authors able to determine whether a participant had type 1 or type 2 diabetes?
Results:
6. It is not clear why Table 1 has been stratified by age groups. The descriptive statistics may be more informative if the table was stratified by prediabetes/diabetes status.
7. Could the authors comment on differences in BMD between glycaemia groups from the descriptive statistics? This is of interest since BMD has been reported to be either not different or higher for individuals with type 2 diabetes.
8. In the Tables, the difference between “left hip fracture” and “right hip fracture” is unclear. Could the authors please explain this in more detail? Additionally, could the authors please explain the difference between “left major osteoporotic fracture” and “right major osteoporotic fracture?”
9. There are a large number of tables in this manuscript. These should be condensed to make the manuscript clearer and easier to read. One suggestion could be to combine similar tables such as Tables 2 and 3, Tables 4-6 and Tables 7-9. Alternatively, since FRAX includes age in the algorithm, it may not be necessary to stratify by age, and all men/women could be presented together in a single table.
Discussion:
10. The authors may need to discuss the effects of using medical records for this study. Is HbA1c measured routinely at the hospital? Or would it be performed only for participants where there was a suspicion of prediabetes or diabetes?
Author Response
Reviewer 2, comment 1:
Thank you for the opportunity to review this manuscript. I have previously reviewed this manuscript for another journal. The authors have responded to some of my previous comments, but not all of them. Please see below for the remaining comments from my previous review:
This is a well written manuscript that presents some interesting results. However, it requires some editing prior to acceptance for publication. In particular, the most interesting message of this manuscript doesn’t stand out, which is the effect of adjusting FRAX using TBS for participants with prediabetes or diabetes. The manuscript can be edited to highlight these observations more clearly.
Abstract:
The results presented in the abstract are unclear. Could the authors please provide some values in the abstract to show the results more clearly?
Response to Reviewer 2, comment 1:
We appreciated the comment from Reviewer 2 and have added numerical results in the abstract.
----------------------------------------------------------------------------------------
Reviewer 2, comment 2:
A participant flow chart would be useful for this study, to show the exclusions and how many participants were included in the final study.
Response to Reviewer 2, comment 2:
We have added a flow chart (Figure 1) in the revised manuscript.
----------------------------------------------------------------------------------------
Reviewer 2, comment 3:
Were there any participants taking anti-fracture medication? Were these included or excluded from the study?
Response to Reviewer 2, comment 3:
We do not have information whether the participants were taking anti-fracture medication. However, we excluded all participants who had a history of fracture.
----------------------------------------------------------------------------------------
Reviewer 2, comment 4:
Additional information is needed about how the “other” information included in the FRAX algorithm was obtained. Would it be that medical records were examined and the appropriate data extracted? Was the data largely complete? Or did some participants have missing data?
Response to Reviewer 2, comment 4:
The “other” information included in the FRAX algorithm was obtained through medical records and questionnaire, which is part of the general health examination. Patients without data on TBS, BMD, HbA1c, and prior fracture history were excluded from the analysis. The other information required for the FRAX calculation was complete.
----------------------------------------------------------------------------------------
Reviewer 2, comment 5:
Were there any participants with type 1 diabetes? Were the authors able to determine whether a participant had type 1 or type 2 diabetes?
Response to Reviewer 2, comment 5:
Our database does not contain information to determine whether a participant had type 1 or type 2 diabetes.
----------------------------------------------------------------------------------------
Reviewer 2, comment 6:
It is not clear why Table 1 has been stratified by age groups. The descriptive statistics may be more informative if the table was stratified by prediabetes/diabetes status.
Response to Reviewer 2, comment 6:
We follow the reviewer’s suggestion and have revised Table 1 with stratification by prediabetes/diabetes status.
----------------------------------------------------------------------------------------
Reviewer 2, comment 7:
Could the authors comment on differences in BMD between glycaemia groups from the descriptive statistics? This is of interest since BMD has been reported to be either not different or higher for individuals with type 2 diabetes.
Response to Reviewer 2, comment 7:
We agree with reviewer that bone mineral density has been reported to be either not different or higher for individuals with type 2 diabetes. A systematic reviews of 47 articles published from January 1950 to October 2010 stated that the majority of articles (26) showed increased, while 13 articles revealed decreased bone mineral density moreover, eight articles revealed normal or no difference in bone mass. [Abdulameer SA, et al. (2012) Osteoporosis and type 2 diabetes mellitus: what do we know, and what we can do?. Patient Preference and Adherence, 6, 435–448.]
In our study, we found a significant higher bone mineral density (in lumbar spine, hip neck femoral, and total hip) in individuals with diabetes. We have added a new Figure 2 to show our results in the revised manuscript.
----------------------------------------------------------------------------------------
Reviewer 2, comment 8:
In the Tables, the difference between “left hip fracture” and “right hip fracture” is unclear. Could the authors please explain this in more detail? Additionally, could the authors please explain the difference between “left major osteoporotic fracture” and “right major osteoporotic fracture?”
Response to Reviewer 2, comment 8:
We thank the reviewer for raising this issue. In our study, we measured the bone mineral density of both the left and right hip and femoral head. We then used these values to estimate the risk of hip fracture and major osteoporotic fracture, separately for the left and right side, based on the FRAX®and TBS-adjusted FRAX®.
The use of bilateral hip scanning is not mandatory in international guidelines for screening of osteoporosis. To improve the clarity of the representation of our data, we have kept only the results for the left side and have also reduced the numbers of table from nine to four. We used the left side data based on the assumption that it is the non-dominant side and that the non-dominant side is less physically active and therefore exposed to less stress and impact, resulting in lower bone mineral density. We have also added a new reference (Petley et al., 2000) to support the use of unilaterial measurements of bone mineral density in the revised manuscript.
----------------------------------------------------------------------------------------
Reviewer 2, comment 9:
There are a large number of tables in this manuscript. These should be condensed to make the manuscript clearer and easier to read. One suggestion could be to combine similar tables such as Tables 2 and 3, Tables 4-6 and Tables 7-9. Alternatively, since FRAX includes age in the algorithm, it may not be necessary to stratify by age, and all men/women could be presented together in a single table.
Response to Reviewer 2, comment 9:
We have reduced the numbers of tables from nine to four by combining Tables 4 to 6 and Tables 7 to 9. We removed the data for the right side of the BMD measurement to improve the clarity of our findings.
----------------------------------------------------------------------------------------
Reviewer 2, comment 10:
The authors may need to discuss the effects of using medical records for this study. Is HbA1c measured routinely at the hospital? Or would it be performed only for participants where there was a suspicion of prediabetes or diabetes?
Response to Reviewer 2, comment 10:
HbA1c is measured routinely for individuals underwent a general health examination in our study hospital.
Reviewer 3 Report
Thank you for a very comprehensive and well-written paper regarding the use of TBS-FRAX and FRAX in Taiwan. A few minor comments for your consideration:
1. In the introduction of the paper, the final three paragraphs [lines 67-114] are very detailed. Some of the details of the results of these studies could be relocated to the discussion section, and summarised more succinctly in the introduction component.
2. Could the authors please provide additional detail for an international audience about the general health exam undertaken by participants. Is this routinely provided for all Taiwanese adults or is it an opt-in process? Further detail would help elucidate some of the potential bias present in the sampling method (which could also be added to the discussion).
3. Please include in the methods a description of how femoral neck BMD was performed for inclusion in FRAX scores. Later in the paper, "left" and "right" fracture risk was indicated in tables, but what this refers to was unclear. My assumption is perhaps the inclusion of left or right femoral neck BMD in the calculation of FRAX, but from my read, I was unsure. Please clarify this in the methods as well. If this is the case, it may also be more prudent to select a single femoral neck score for inclusion in analyses and present only one set of this data, as it did convolute the paper somewhat.
4. Similarly, there are overall a large number of tables and data supplied in this paper, which somewhat muddies the message. I would recommend moving Tables 4-9 to supplementary data, and instead providing a table for overall differences between glycaemic status for men and women, that is, without the age categorisation. This would also help reduce the large number of underpowered analyses where t-tests could not be applied due to 0 cell fields.
5. Finally, I would broadly recommend expand the discussion slightly to include commentary on why you think the current paper is similar and different to other published studies. This could still be done with the consideration that the analyses are not directly comparable due to differences in approach.
6. Overall, a very well done paper! Congratulations!
Author Response
Reviewer 3, comment 1:
Thank you for a very comprehensive and well-written paper regarding the use of TBS-FRAX and FRAX in Taiwan. A few minor comments for your consideration:
In the introduction of the paper, the final three paragraphs [lines 67-114] are very detailed. Some of the details of the results of these studies could be relocated to the discussion section, and summarised more succinctly in the introduction component.
Response to Reviewer 3, comment 1:
We appreciate the reviewer’s suggestion. The paragraph from line 72 to 86 introduces the background why TBS can be added to improve FRAX® assessments, and that there are some inconsistencies in the results from various studies. The paragraph from line 87 to 101 provides the rationale of why we selected ³3% for hip fracture and ³20% for major osteoporotic fracture as a cutoff value for intervention in this study. The text from line 102 to 114 gives the background information on why glucose regulation status was explored in this study. We believe this background information is useful for the general readers to obtain in the introduction, and therefore, we hope the reviewer will agree with us that these three paragraphs can remain in the Introduction section.
----------------------------------------------------------------------------------------
Reviewer 3, comment 2:
Could the authors please provide additional detail for an international audience about the general health exam undertaken by participants. Is this routinely provided for all Taiwanese adults or is it an opt-in process? Further detail would help elucidate some of the potential bias present in the sampling method (which could also be added to the discussion).
Response to Reviewer 3, comment 2:
In Taiwan, all residents are eligible to receive a free comprehensive health examination once every three years for those who are 40 to 64 years old, and once a year for those aged 65 and older. We have added this information in the Methods section (line 125-127). Therefore, our sample should not be biased by financial issues.
----------------------------------------------------------------------------------------
Reviewer 3, comment 3:
Please include in the methods a description of how femoral neck BMD was performed for inclusion in FRAX scores. Later in the paper, "left" and "right" fracture risk was indicated in tables, but what this refers to was unclear. My assumption is perhaps the inclusion of left or right femoral neck BMD in the calculation of FRAX, but from my read, I was unsure. Please clarify this in the methods as well. If this is the case, it may also be more prudent to select a single femoral neck score for inclusion in analyses and present only one set of this data, as it did convolute the paper somewhat.
Response to Reviewer 3, comment 3:
We greatly appreciate the reviewer for this suggestion to help us focusing on the main findings of this study. We follow the suggestion and have kept only the results for the left side and the numbers of table are now reduced from nine to four. The main findings and conclusion are indeed the same based on unilaterial (only the left side) data.
----------------------------------------------------------------------------------------
Reviewer 3, comment 4:
Similarly, there are overall a large number of tables and data supplied in this paper, which somewhat muddies the message. I would recommend moving Tables 4-9 to supplementary data, and instead providing a table for overall differences between glycaemic status for men and women, that is, without the age categorisation. This would also help reduce the large number of underpowered analyses where t-tests could not be applied due to 0 cell fields.
Response to Reviewer 3, comment 4:
To improve the clarity of the representation of our data, we have kept only the results for the left side and have also reduced the numbers of table from nine to four. We used the left side data based on the assumption that it is the non-dominant side and that the non-dominant side is less physically active and therefore exposed to less stress and impact, resulting in lower bone mineral density. We have also added a new reference (Petley et al., 2000) to support the use of unilaterial measurements of bone mineral density in the revised manuscript.
----------------------------------------------------------------------------------------
Reviewer 3, comment 5:
Finally, I would broadly recommend expand the discussion slightly to include commentary on why you think the current paper is similar and different to other published studies. This could still be done with the consideration that the analyses are not directly comparable due to differences in approach.
Response to Reviewer 3, comment 5:
As mentioned at the end of Introduction section, previous studies have not evaluated the consistency between the FRAX® and TBS-FRAX® in identifying individuals qualified for osteoporosis intervention, especially separately in men and women of different age groups, as well as in individuals with normoglycemia, prediabetes, and diabetes. We follow the suggestion by the reviewer and added the following sentences at the end of the Discussion section “Third, the present study provided comprehensive comparisons of the risk of major osteoporotic fracture and hip fracture assessed by FRAX® or TBS-FRAX® separately for men and women of different age groups, as well as in individuals with normoglycemia, prediabetes, and diabetes.”. To the best of our knowledge, existing studies have not provided similar comparisons with stratification by age group, sex, and glucose regulation status.
----------------------------------------------------------------------------------------
Reviewer 3, comment 6:
Overall, a very well done paper! Congratulations!
Response to Reviewer 3, comment 6:
We thank the reviewer for spending the time to review our manuscript and for providing the valuable comments.
Round 2
Reviewer 2 Report
The authors have done an excellent job of revising their manuscript. The Results section in particular is now much clearer. There were just two minor edits remaining that could help strengthen the manuscript.
1) Regarding the flow chart: It is great to see that the authors have added a flow chart to the manuscript. However, the flow chart includes only one numerical value (8098). Could the authors please add additional values such as the number of medical records that were searched and the number of participants that were excluded? If these numbers are not known, could the authors please mention this in the Methods section of the manuscript? [Reviewer 2, comment 2]
2) Regarding the comments where the authors did not have enough information to determine anti-fracture medication use and type 1 diabetes, could the authors please mention this in their Discussion section as a limitation? This information is important for FRAX and it would be good to acknowledge the limitations. [Reviewer 2, comments 3 & 5]
Author Response
Reviewer 2, comment 1:
The authors have done an excellent job of revising their manuscript. The Results section in particular is now much clearer. There were just two minor edits remaining that could help strengthen the manuscript.
1) Regarding the flow chart: It is great to see that the authors have added a flow chart to the manuscript. However, the flow chart includes only one numerical value (8098). Could the authors please add additional values such as the number of medical records that were searched and the number of participants that were excluded? If these numbers are not known, could the authors please mention this in the Methods section of the manuscript? [Reviewer 2, comment 2]
Response to Reviewer 2, comment 1:
We have revised Figure 1 and added the numbers of individuals excluded for various reasons.
----------------------------------------------------------------------------------------
Reviewer 2, comment 2:
2) Regarding the comments where the authors did not have enough information to determine anti-fracture medication use and type 1 diabetes, could the authors please mention this in their Discussion section as a limitation? This information is important for FRAX and it would be good to acknowledge the limitations. [Reviewer 2, comments 3 & 5]
Response to Reviewer 2, comment 2:
We greatly appreciate the comment from the reviewer and added this point in the limitation section.